# Proteomics Analysis of Plasma Membrane Fractions of the Root, Leaf, and Flower of Rice

**DOI:** 10.3390/ijms21196988

**Published:** 2020-09-23

**Authors:** Yukimoto Iwasaki, Takafumi Itoh, Yusuke Hagi, Sakura Matsuta, Aki Nishiyama, Genki Chaya, Yuki Kobayashi, Kotaro Miura, Setsuko Komatsu

**Affiliations:** 1Department of Bioscience and Biotechnology, Fukui Prefectural University, Fukui 910-1195, Japan; ito-t@fpu.ac.jp (T.I.); hagiyusuke@fpu.ac.jp (Y.H.); matsutasakura@fpu.ac.jp (S.M.); nishiyamaaki@fpu.ac.jp (A.N.); s1873012@g.fpu.ac.jp (G.C.); s1721014@g.fpu.ac.jp (Y.K.); miura-K@fpu.ac.jp (K.M.); 2Department of Environmental and Food Sciences, Fukui University of Technology, Fukui 910-8505, Japan

**Keywords:** heterotrimeric G protein, proteomics, rice, plasma membrane, organ specificity

## Abstract

The plasma membrane regulates biological processes such as ion transport, signal transduction, endocytosis, and cell differentiation/proliferation. To understand the functional characteristics and organ specificity of plasma membranes, plasma membrane protein fractions from rice root, etiolated leaf, green leaf, developing leaf sheath, and flower were analyzed by proteomics. Among the proteins identified, 511 were commonly accumulated in the five organs, whereas 270, 132, 359, 146, and 149 proteins were specifically accumulated in the root, etiolated leaf, green leaf, developing leaf sheath, and developing flower, respectively. The principle component analysis revealed that the functions of the plasma membrane in the root was different from those of green and etiolated leaves and that the plasma membrane protein composition of the leaf sheath was similar to that of the flower, but not that of the green leaf. Functional classification revealed that the root plasma membrane has more transport-related proteins than the leaf plasma membrane. Furthermore, the leaf sheath and flower plasma membranes were found to be richer in proteins involved in signaling and cell function than the green leaf plasma membrane. To validate the proteomics data, immunoblot analysis was carried out, focusing on four heterotrimeric G protein subunits, Gα, Gβ, Gγ1, and Gγ2. All subunits could be detected by both methods and, in particular, Gγ1 and Gγ2 required concentration by immunoprecipitation for mass spectrometry detection.

## 1. Introduction

The plasma membrane, which separates the intercellular components from the extracellular environment, plays fundamental roles in cell activities. As the plasma membrane is the first site of extracellular biotic or abiotic sensing, an understanding of the dynamics of plasma membrane proteins facilitates the development of new strategies for stress tolerance and resistance in crops [1]. The plasma membrane regulates biological processes such as ion transport, endocytosis, cell differentiation/proliferation, and signal transduction [2]. Functional protein localization is a complex process as some membrane proteins are tightly associated with the dual lipid core, whereas others are loosely and reversibly associated [3]. While the purification of high-quality plasma membranes is difficult, it is important to analyze the mechanisms underlying plant responses to environmental cues. 

The development of technologies for plasma membrane protein extraction has allowed the profiling of proteins in the entire plant plasma membranes [4]. Proteomics can be used to explore and discover the mechanisms that regulate biological activities and physiological phenomena [5]. Proteomics analyses have provided new insights that can be used to explore the potential of enhancing stress tolerance [6]. High-throughput proteomics studies can not only reveal the mechanisms of stress-related metabolic responses, but also reflect the specificity of different stress factors [7]. Protein profiling by proteomics can reveal the changes in metabolic processes at different plant developmental stages and elucidate systemic protective responses that are initiated in plant plasma membranes exposed to environmental stress.

The global demand for staple crops is expected to increase by 60% from 2010 to 2050 [8]. Rice, wheat, and maize are the three major cereal crops worldwide. Rice is a model crop and the first cereal to be sequenced [9], and extensive technological platforms and analytical tools have been established for functional genomic research in rice. The names, websites, references, working states, and number of citations for various rice databases and tools have been reviewed by Hong et al. [10]. Along with accumulating wheat and maize genomic data, advances in systematic approaches in rice could facilitate efforts to improve agronomic traits of other crops.

Transcriptome databases provide genome-wide expression profiles. As a part of the PlantExpress database [11], OryzaExpress provides gene expression data of 1206 samples in 34 experimental series [12]. While information about organ-specific expression is included in transcriptome data, subcellular localization of protein can only be predicted using bioinformatic analysis. In the present study, proteins enriched in the plasma membrane fractions were analyzed using proteomics approaches to enhance our understanding of the function and characteristics of plasma membranes of different organs, including the root, etiolated seedling, green leaf, leaf sheath, and flower in rice. 

## 2. Results

### 2.1. Purification of Rice Organ Plasma Membrane and Evaluation of Purity

To understand and compare the functions of the plasma membranes in various rice organs, plasma membrane fractions of the root, etiolated seedling, green leaf, leaf sheath, and flower were prepared in triplicate, using an aqueous two-polymer phase system; the five organs were sampled from growing rice plants (Figure 1). To determine how the proportion of the plasma membrane increased in the membrane fraction, immunoblotting with an antibody against the plasma membrane marker protein aquaporin (OsPIP1s) was carried out. The aquaporin concentration in the plasma membrane fractions was compared with that in the crude microsomal fractions (Figure 2). 

### 2.2. Principle Component Analysis (PCA) of Plasma Membrane Proteins from Different Organs of Rice

Proteins in the plasma membrane fractions were separated by sodium dodecyl sulfatepolyacrylamide gel electrophoresis (SDS-PAGE) and the proteins in the gels were reduced, alkylated, and digested with trypsin. The digested proteins were analyzed by liquid chromatography tandem mass spectrometry (LC-MS/MS) (Figure 1). The proteome data, comprising 2667 proteins from the 15 samples (triplicate samples of the five organs), were subjected to PCA, which revealed that the protein expression patterns varied with the organ (Figure 3). Based on the MS data of triplicate samples, the identified proteins were highly reliable and were thus used for further analyses. In total, more than 1000 proteins were identified in the plasma membrane fractions of the root (1181), etiolated leaf (1314), green leaf (1400), leaf sheath (1369), and flower (1279), respectively (Appendix A).

### 2.3. Overall Aspects of Organ-Specific and Common Accumulated Rice Proteins in the Plasma Membrane Fractions

A Venn diagram was generated to show the number of common and organ-specific plasma membrane proteins among proteins from the root (1181), etiolated leaf (1314), green leaf (1400), leaf sheath (1369), and flower (1279), respectively (Figure 4). In total, 511 proteins were commonly accumulated in all organs evaluated, whereas there were specifically accumulated proteins in the root (270), etiolated leaf (132), green leaf (359), leaf sheath (146), and flower (149), respectively. In the Gene Ontology analysis, among the 511 common proteins, 207 proteins were categorized as plasma membrane proteins (Figure 4 and Appendix A). In the UniProtKB analysis, 445 proteins were categorized as plasma membrane proteins specific to the root (134), etiolated seedling (64), green leaf (119), leaf sheath (59), and flower (69), respectively (Appendix A). 

### 2.4. Proteins in the Plasma Membrane Fractions in the Root, Etiolated Leaf, and Green Leaf 

From the PCA plot (Figure 3), it was apparent that the plasma membrane proteins in the root, as the underground plant part, were different from those in etiolated and green leaves, which are aerial parts. Therefore, the proteins in the plasma membrane fraction of root were compared with those of etiolated and green leaves. In total, 701 proteins were commonly accumulated in the root, etiolated leaf, and green leaf, whereas 944 (418 + 251 + 275) proteins were specifically accumulated in the green and etiolated leaves, and 303 were root specific (Figure 5).

The proteins in these three organs were functionally categorized into the following categories using the MapMan bin codes: transport, signaling, cell, stress, lipid metabolism, redox, cell wall, hormone metabolism, secondary metabolism, and development. The transport and signaling categories were the most enriched in all three organs (Figure 5). The root plasma membrane had some specific transport-related proteins compared with the green and etiolated leaf plasma membrane. 

### 2.5. Specific Transport-Related Proteins in the Root Plasma Membrane 

By focusing on transport proteins, their accumulation profiles in the root (R), green leaf (GL) and etiolated leaf (EL) were analyzed using the MapMan bin analysis (Table 1 and Appendix A). A total of 210 proteins were divided into seven groups—1 group of proteins common among the three organs (R/EL/GL), three groups of proteins that accumulated in two organs (R/EL, R/GL, or EL/GL), and three groups of proteins showing organ specificity (R, EL, or GL)—according to the MapMan bin codes. There were 59 root-specific proteins, which was the maximum number found among the seven groups, including four ammonium transporters, six metal transporters, and seven aquaporin as the most predominant proteins (Table 1 and Appendix A). 

### 2.6. Proteins in the Plasma Membrane Fractions of the Leaf Sheath and Flower

The PCA plot (Figure 3) indicates that the plasma membrane protein compositions of the leaf sheath and flower are similar. Therefore, the plasma membrane proteins in the leaf sheath and flower were compared with those in the green leaf as a reference of a mature organ. In total, 695 proteins were commonly accumulated in leaf sheath, flower, and green leaf (Figure 6). The proteins were functionally categorized into transport, signaling, cell, stress, lipid metabolism, redox, cell wall, hormone metabolism, secondary metabolism, and development, using the MapMan bin codes (Figure 6). The highest number of proteins was found in the following categories for all three organs: signaling, cell, and transport. The number of proteins in the signaling and cell categories was substantially higher in the leaf sheath and flower than in the green leaf, whereas that in the transport category was lower in the leaf sheath and flower than in the green leaf. Thus, the plasma membranes of the flower and leaf sheath have some similar characteristics compared with those of the green leaf. 

### 2.7. Specific Proteins Related to Cell and Signaling in the Plasma Membrane of the Leaf Sheath and Flower

The proteins in category “cell”, commonly detected in the leaf sheath and flower, were compared with those in the green leaf as a reference (Table 2 and Appendix A). A total of 162 proteins were further categorized into the following groups, using the MapMan bin codes: organization, division, cycle, and vesicle transport. These 162 proteins were divided into seven groups, namely, one group of proteins detected in all three organs (R/EL/GL), three groups of proteins detected in two organs (R/EL, R/GL, or EL/GL), and three groups of proteins that were organ specific (R, EL, or GL). Among the seven groups, F/LS/GL and F/LS had the highest number of proteins, that is, 45 and 43, respectively (Table 2). In category “cell organization”, 24 proteins, including actin, actin-depolymerizing factor, tubulin, annexin, kinesin, and dynein, were detected in the F/LS/GL group; and 23 proteins, including actin, tubulin, kinesin, and dynein were identified in the F/LS group. Therefore, actin, tubulin, kinesin, and annexin were detected in both F/LS and the F/LS/GL groups. 

In category “cell vesicle transport”, 16 proteins, including clathrin, t-SNARE, exocyst complex component, and coatomer, were detected in F/LS/GL; and 17 proteins, including coatomer, exocyst complex component, adaptin, and Golgi SNAP receptor, were identified in F/LS. 

In category “signaling”, 265 proteins (integrated value of subtotal of Table 3) were categorized into the following groups, using MapMan bin codes: sugar/nutrient physiology, receptor kinase, calcium, phosphoinositide, G-protein, MAP kinase, 14-3-3, lipids, light, and specified. The 265 proteins were also grouped into seven groups as described above (F/LS/GL, F/LS, F/GL, LS/GL, F, LS, GL); there were 99 proteins in F/LS/GL and 47 in F/LS (Table 3). 

In total, 36 receptor kinases, including two brassinosteroid-insensitive 1-associated receptor kinase 1 (BAK1; accession Nos: Q75I95 and B8BB68), were found in the F/LS/GL group and 24 receptor kinases were found in the F/LS group in the UniProt database. (Table 3 and Appendix A). In total, 33 G-protein-related proteins, including heterotrimeric G protein α subunit (accession no: A2Y3B5) and β subunit (accession no: Q40687), were accumulated in the F/LS/GL group and 10 G-protein-related proteins, including heterotrimeric Gγ2 subunit (accession no: Q6YXX9), were accumulated in the F/LS group. The Gγ2 subunit was detected in the plasma membranes of flower and leaf sheath, but not in that of green leaf (Appendix A).

### 2.8. ATPase and Signal Transduction Proteins in Rice Plasma Membrane

Plasma membrane ATPases (p-type ATPases) transport numerous species of ions such as H^+^, Na^+^, K^+^, and Ca^2+^ in both directions [13]. Forty-three members of the p-type ATPase superfamily have been identified in rice based on genomic data [14]. In the present study, 31 p-type ATPases were identified (Appendix A) and the expression profiles of 28 ATPases detected in the root, etiolated leaf, and green leaf are summarized in Appendix A.

Receptor kinases are also dominant signaling proteins in the plasma membrane, because 265 proteins were categorized as “signaling”, approximately half of which (122 proteins) were further categorized as receptor kinases (Table 3). With regard to hormone signal transduction, two BAK1 were detected in the F/LS/GL group (Appendix A. The BRI1 brassinosteroid receptor, categorized under hormone metabolism, and not signaling, was detected in the flower, leaf sheath, and green leaf (Appendix A). Thus, the proteomics analysis successfully detected BRI1 and BAK1 as brassinosteroid signaling molecules. We detected 24 specific receptor kinases accumulated specifically in the F/LS group, and the ligands of the receptor kinases are not clear.

G-protein-related proteins, 63 in total, were the second-most dominant group in signaling, and 33 proteins were accumulated in the F/LS/GL group. Among the 33 proteins, Gα and Gβ were detected in the flower, leaf sheath, and green leaf (Table 3). Gγ2 was detected in the flower and leaf sheath, but not the green leaf.

### 2.9. Verification of the Proteome Data by Immunoblotting 

To verify the proteome data of plasma membrane proteins, we performed immunoblotting, focusing on the rice heterotrimeric G protein subunits. Using the proteomics approach, Gα and Gβ were detected in the flower, leaf sheath, and green leaf; and Gγ2 was detected in the flower and leaf sheath, but not in the green leaf (Table 3 and Appendix A). We detected 15 Gα and 12 Gβ fragments in the plasma membrane fraction (*p* < 0.05) (Appendix A, respectively). 

By immunoblotting, Gα, Gβ, Gγ1, and Gγ2 were detected in the five organs examined (Figure 7). To identify the Gγ1 subunit by MS and enhance the accuracy of the MS data for the Gγ2 subunit, both Gγ1 and Gγ2 were concentrated by immunoprecipitation (IP) (Figure 8), and then analyzed by LC-MS/MS. We detected 5 Gγ1 and 8 Gγ2 fragments in the immunoprecipitants (*p* < 0.05) (Appendix A, respectively). In Figure 9, these are underlined in the full-length Gγ1 and Gγ2 amino acid sequences. Thus, the Gγ1 subunit was identified by MS analysis, and MS data accuracy for the Gγ2 subunit was increased. 15 Gα and 12 Gβ fragments (Appendix A, respectively) are also underlined in the full-length Gα and Gβ amino acid sequences (Figure 9).

## 3. Discussion

The isolation and purification of subcellular organelles enable the identification and functional characterization of specific proteins in each compartment. However, the accuracy of such results depends on the purity of the targeted organelle, and it is influenced by the degree of protein enrichment and extent of contamination from other subcellular fractions [15]. In highly pure plasma membranes, the ATPase activity can be assayed in the presence of specific inhibitors; Na_3_VO_4_, KNO_3_, and NaN_3_ have been used as inhibitors of plasma membrane, tonoplast, and mitochondrial membrane enzyme activities, respectively [16,17]. In the present study, the accumulation of the plasma membrane marker OsPIP1s was assessed to evaluate the purity of plasma membrane. The results indicated that the plasma membrane fractions of all organs were purified at the immunoblotting level, and the fractions were clearly clustered in the PAC plot. 

Plasma membrane proteins have been identified in *Arabidopsis* [1], *Medicago* [18], rice [19,20], maize [21], and oat/rye [22]. In *Arabidopsis*, 238 plasma-membrane proteins related to transport, signal transduction, membrane trafficking, and stress responses have been identified in the leaves and petioles [1], and 188 proteins related to transport, signal detection, cell wall, and membrane trafficking have been identified in the roots and leaves [23]. In the roots of *Medicago*, 96 plasma membrane proteins related to arbuscular–mycorrhizal symbiosis, including flotillin-like proteins, which support membrane trafficking during mycorrhizal establishment, have been identified [18]. In maize, 204 and 251 plasma membrane proteins including membrane-bound redox proteins have been identified under iron-depleted and -rich conditions, respectively [21]. The expression of P-type ATPase and aquaporins was induced in both in oat and rye during cold acclimation, and that of heat shock protein 70 was increased in oat [22]. These results indicate that plasma membrane proteins are involved in transport, membrane trafficking, and redox homeostasis in plants exposed to biotic and abiotic stresses. In this study, the plasma membrane proteins identified in the five organs were classified into the following categories: transport, signaling, cell (cell metabolism), stress lipid metabolism, redox, cell wall, hormone metabolism, secondary metabolism, and development. These categories were consistent with those reported in previous studies. 

Roth et al. [24] purified the plasma membrane and identified 2036 proteins from untreated rice and 3550 proteins including 591 fungal proteins from inoculated rice as rice membrane proteins. Using these proteins, peri-arbuscular membrane-specific arbuscular receptor-like kinase 1 for sustained arbuscular mycorrhizal symbiosis was detected. Although the number of plasma membrane proteins identified in rice root in our study was low, many proteins were the same as plasma membrane proteins identified by Roth et al. [24]. In rice leaves, 3049 proteins have been previously identified [20], 1172 of which exist only in the plasma membrane fraction and 1877 in the cytosolic fraction. In the present study, 1369 proteins were identified as plasma membrane proteins in the leaf sheath of rice. In addition, proteomic studies on the microdomain structure of plasma membrane in oat, rye [22], and rice [25] could enhance our understanding of plasma membrane functions in plants. 

Considering the available genomic, microarray, and proteomic data, posttranslational regulation is also essential for the regulation of physiological activities. It has been suggested that posttranslational regulation of plasma membrane H^+^-ATPase is important in evolutionary and physiological contexts [26]. When constitutively activated mutant forms of plasma membrane H^+^-ATPase are expressed throughout the plant, developmental problems arise that likely result from active ATPase expression in cells where it is not normally expressed [27]. There is strong genetic evidence for specific physiological roles of plasma membrane H^+^-ATPases, particularly in the stomata [28]. The expression of an activated form of a plasma membrane-specific H^+^-ATPase isoform from its natural promoter resulted in the constitutive swelling of stomatal guard cells [26]. 

The primary functions of the root system are anchorage of the plant and absorption of water and minerals. In this study, 21 major plasma membrane intrinsic proteins (PIPs), including aquaporins, were detected in the five organs, and 16 aquaporins were accumulated in the root plasma membrane. Plant aquaporins exist in numerous isoforms. Thirty-five homologs belonging to four homology subclasses have been identified in *Arabidopsis*: PIPs, with 13 isoforms, which are further subdivided into the PIP1 and PIP2 subgroups, and tonoplast intrinsic proteins, with 10 homologs, are the most abundant aquaporins in the plasma membrane and tonoplast, respectively [29,30]. Using the interactome analysis, Bellati et al. [31] revealed that PIPs are involved in a wide range of transport activities and provide novel insights into the regulation of PIP cellular trafficking by osmotic and oxidative treatments. Transporters of nitrate, ammonium, sulfate, metal, and anions as well as ABC transporters are also abundant in rice root. Our findings support the physiological roles of rice root at the proteome level.

In category “cell organization”, actin, tubulin, kinesin, and annexin were detected in both F/LS and the F/LS/GL groups. The proteins in the F/LS and F/LS/GL groups presented similar functions, but were encoded by different genes. Different gene family members can be found in different organs. In the vesicle transport category, proteins found in the F/LS group were largely different from those in the F/LS/GL group. This result indicated that a different set of proteins and enzymes is involved in vesicle transport in the flower and leaf sheath.

Using the proteomic approach, Gα and Gβ were detected in the flower, leaf sheath, and green leaf. Gγ2 was detected in the flower and leaf sheath, but not in the green leaf. Rice has 1 Gα, 1 Gβ, and 5 Gγ subunits. Our results suggest that the Gα and Gβ subunits are house-keeping types, whereas the Gγ subunit may be organ-specific.

As many proteins were identified as candidate plasma membrane proteins using the proteomics approach (Appendix A), we conducted a verification experiment by immunoblotting, focusing on the rice heterotrimeric G protein subunits. The heterotrimeric G protein complex consists of three subunits, Gα, Gβ, and Gγ [32]. The Gα subunit with GDP can complex with Gβγ dimer to form the heterotrimer. Rice has one Gα subunit gene [33], one Gβ subunit gene [34] (Ishikawa et al. 1996), and five Gγ subunit genes (reviewed in a previous study [32]). Gγ1 is a canonical subunit and Gγ2 is a monocot-specific subunit [35]. Gγ3, Gγ4, and Gγ5 are chimeric proteins with cysteine-rich residues [35]. The Gγ3 and Gγ4 genes correspond to the rice genes *GS3* [36] and *DEP1* [37], respectively, which have been used in breeding [38]. 

Previously, we detected Gα, Gβ, Gγ1, and Gγ2 in the plasma membrane fraction of green leaf by immunoblotting using subunit-specific antibodies [39]. In addition, trypsin-digested fragments of Gα and Gβ have been detected by LS-MS/MS using the Gα and Gβ IP products [40]. In this study, Gα and Gβ were detected in the plasma membrane fractions by LS-MS/MS, without IP, likely because of the high sensitivity of modern equipment. Gγ2 was detected by LS-MS/MS in the leaf sheath and flower, but not in the green leaf. Gγ1 was not detected by LS-MS/MS in any of the organs. Thus, Gγ2 might be present at a level below the LS-MS/MS detection limit in the green leaf and Gγ1 might be present at levels below the threshold in all organs, as we did not identify both subunits immunologically. Therefore, we immunologically concentrated the Gγ1 and Gγ2 subunits, and then detected them by LC-MS/MS. Gγ3 and Gγ4 in rice have been identified by IP [41,42]. Similar to Gγ1 and Gγ2, Gγ3, and Gγ4 are difficult to detect in trypsin-digested plasma membrane fractions by LC-MS/MS. These results suggest that at present, the sensitivity of immunoblot analysis is substantially higher than that of MS analysis.

Finally, the PCA results of the total plasma membrane proteome data of the five organs indicated that each organ has characteristic plasma membrane proteins. Western blotting analysis, focusing on the heterotrimeric G protein subunits, revealed the presence of the Gα and Gβ subunits. Our proteome data of the five organs will help future plant science research. 

## 4. Materials and Methods 

### 4.1. Plant Materials

Rice (*Oryza sativa* L.) cultivar Nipponbare was used in this study. We sampled root and etiolated leaf from plants grown for 1 week in dark at 28 °C and green leaf from plants grown for one week under continuous light (50,000 lux) at 28 °C. Approximately 2000 tillers were harvested from the paddy field and dissected in our lab each day. The leaf sheath length of the youngest leaf of the tiller varied from 0 to 20 cm. We collected leaf sheath of length 1–5 cm, and it accounted for approximately 10% of the dissected tillers. The weight of 1–5-cm long leaf sheath obtained from 200 tillers was approximately 4 g fresh weight. From 4 g of leaf sheath, 200–300 μg of plasma membrane was obtained. The sampling of flower was similar to that of leaf sheath. Approximately 2000 tillers were harvested from the paddy field and dissected in the lab each day. The tillers that contained 1–5 cm flowers accounted for approximately 10% of the total tillers harvested; 200 tillers were used. The weight of 1–5-cm long flowers from 200 tillers was approximately 4 g fresh weight. From 4 g of flowers, 200–300 μg of plasma membrane was obtained. These extracts were used as samples for biological triplicates (Figure 1).

### 4.2. Preparation of Microsomal and Plasma Membrane Fractions 

All procedures for plasma membrane preparation were performed at 4 °C. Crude microsomal fractions were prepared from the root, etiolated leaf, green leaf, leaf sheath, and flower, as described previously [39]. The plasma membrane fractions were purified from the crude microsomal fraction using an aqueous two-polymer phase system [43].

### 4.3. SDS-PAGE

Electrophoresis was carried out on 12.5% and 10–20% PA gels containing 0.1% SDS, as described previously [44]. 

### 4.4. Immunoblotting

Proteins (5 μg) were separated by SDS-PAGE. Chemi-Lumi One Markers (Nacalai Tesque, Kyoto, Japan) were used as molecular weight markers. The separated proteins were transferred onto a polyvinylidene difluoride membrane (Millipore, Darmstadt, Germany). After blocking, the membrane was cross-reacted with antibodies, including the anti-OsPIP1s antibody (Operon Biotechnologies, Huntsville, AL, USA) and anti-G protein subunit antibodies. ECL peroxidase-labeled anti-rabbit secondary antibody was used as the secondary antibody (GE Healthcare, Little Chalfont, UK). Immobilon Western chemiluminescent HRP substrate (Millipore) was used for detection. The chemiluminescent signal was measured using the Fusion SL instrument (M&S Instruments, Orpington, UK).

### 4.5. IP

Affinity-purified anti-Gγ1 and anti-Gγ2 antibodies (50 μg) were bound to 50 mg of Protein A-bound magnetic beads (Millipore). After three washes with phosphate-buffered saline, the antibodies and Protein A were cross-linked with dimethyl pimelimidate dihydrochloride. The conditions for cross linking were according to the manufacturer’s protocols. After quenching the magnetic cross-linked beads with the antibodies, 0.1 mL of 10% SDS was added to 0.9 mL of plasma membrane fraction (1 mg protein/10 mg SDS/mL) and denatured at 90 °C for 5 min. After diluting the solubilized fraction with 10 mL of Tris-buffered saline (TBS) containing 1% Tween-20, the magnetic beads cross linked with 50 μg of the subunit specific antibodies were added. After incubation at 4 °C overnight, the magnetic beads were collected into a 1.5 mL tube and washed thrice with 0.5 mL of TBS containing 0.1% Tween-20 and 0.5 mL of TBS each time. The proteins were eluted from the beads using 40 μL of dissociation buffer (Bio-Rad, Hercules, CA, USA) without a reducing agent. Each eluate (5 μL) was subjected to SDS-PAGE and gel cuttings were used for LC-MS/MS.

### 4.6. Sample Preparation for LC-MS/MS 

For LC-MS/MS, protein extracts (20 μg) were separated by SDS-PAGE. The 3-cm long gel was divided into 10 parts according to the molecular weight marker (Precision Plus Protein Kaleidoscope; Bio-Rad), without staining. Gel pieces were resuspended in 50 mM NH_4_HCO_3_, reduced with 50 mM dithiothreitol at 56 °C for 30 min, and alkylated with 50 mM iodoacetamide at 37 °C in the dark for 30 min. Alkylated proteins in the gels were digested with 10 μg/mL trypsin solution (Promega, Madison, WI, USA) at 37 °C for 16 h. The peptides were concentrated and suspended in 0.1% formic acid.

### 4.7. Protein Identification by NanoLC-MS/MS

The peptides were loaded onto an LC system (EASY-nLC 1000; Thermo Fisher Scientific, San Jose, CA, USA) equipped with a trap column (EASY-Column, C18-A1 5 µm, 100 µm ID × 20 mm; Thermo Fisher Scientific), equilibrated with 0.1% formic acid, and eluted with a linear acetonitrile gradient (0–50%) in 0.1% formic acid at a flow rate of 200 nL/min. The eluted peptides were loaded and separated on a C18 capillary tip column (75 µm ID × 120 mm; Nikkyo Technos, Tokyo, Japan) with a spray voltage of 1.5 kV. The peptide ions were detected by MS (LTQ Orbitrap Elite MS; Thermo Fisher Scientific) in the data-dependent acquisition mode with Xcalibur software (version 2.2; Thermo Fisher Scientific). Full-scan mass spectra were acquired over 350–2000 *m*/*z*, at a resolution of 60,000. The 10 most intense precursor ions were selected for collision-induced fragmentation in the linear ion trap, at a normalized collision energy of 35%. Dynamic exclusion was employed within 90 s to prevent repetitive peptide selection.

### 4.8. MS Data Analysis

Protein identification was performed using SEQUEST HT program implemented in Proteome Discoverer software (version 2.4; Thermo Fisher Scientific) and an in-house database of rice amino acid sequences (UniProt database, 423,016 unreviewed sequences downloaded on 28 October 2019). For both searches, cysteine carbamidomethylation was set as a fixed modification and methionine oxidation was set as a variable modification. Trypsin was specified as the proteolytic enzyme and one missed cleavage was allowed. The peptide mass tolerance was set at 10 ppm, fragment mass tolerance was set at 0.8 Da, and peptide charges were set at +2, +3, and +4. Automatic decoy database search was performed as a part of the search. SEQUEST results were filtered with the Percolator function to improve the accuracy and sensitivity of peptide identification. The minimum requirement for the identification of a protein was two matched peptides. Significant changes in the abundance of proteins between samples were determined using the *t*-test (*p* < 0.05). 

### 4.9. PCA

Raw MS data files were analyzed using MaxQuant (version 1.6) [45] for protein identification and label-free quantification using in-house standard parameters. Notably, the data were normalized based on the assumption that the majority of proteins did not differ between the samples. Statistical analysis was performed using an in-house-generated R script based on the ProteinGroup.txt file. First, contaminant proteins, reverse sequences, and proteins identified “only by site” were filtered. In addition, proteins that were identified only by a single peptide and proteins not identified/quantified consistently in the same condition were removed. The LFQ data were converted to the log2 scale, samples were grouped based on organs, and missing values were input using the “Missing not At Random” (MNAR) method, which uses random draws from a left-shifted Gaussian distribution of 1.8 standard deviations separated by a width of 0.3. Protein-wise linear models combined with empirical Bayes statistics were used for differential expression analyses. The limma package in R was used to generate a list of differentially expressed proteins for each pairwise comparison. Adjusted *p* < 0.05 (Benjamini–Hochberg method) along with a |log2 fold-change| of 1 were applied to determine significantly differentially regulated proteins in each pairwise comparison. 

### 4.10. Functional Predictions

Protein functions were categorized using the MapMan bin codes [46]. Assignment of plasma membrane proteins was performed using the UniProKB system [47].

### 4.11. Accession Codes

For MS data, RAW data, peak lists, and result files have been deposited in the ProteomeXchange Consortium [48] via the jPOST [49] partner repository under the following dataset identifiers: PXD018201 for root, PXD018197 for etiolated leaf, PXD018200 for green leaf, PXD018198 for developing leaf sheath, and PXD018199 for flower.

## 5. Conclusions

To understand the functional characteristics and organ specificity of plasma membranes, proteins in the plasma membrane fractions of root, etiolated leaf, green leaf, leaf sheath, and flower were analyzed by proteomics. The main findings of this study are as follows: (i) The p-ATPase, ammonium transporter, metal transporter, and aquaporin were specifically identified in the root plasma membrane; (ii) several signal transduction proteins, including receptor kinase, BRI1, BAK1, heterotrimeric G protein subunits Gα, Gβ, and Gγ2 were identified specifically in the leaf sheath and flower plasma membranes; (iii) the data generated in this study provide useful information about many important proteins and enzymes, such as Gα, Gβ and Gγ2; however, our database does not contain minor proteins and enzymes, such as Gγ1; and (iv) many proteins identified in this study were encoded by multigene families. Some were accumulated in all organs examined, whereas others were organ specific. Thus, this study indicated that some proteins and enzymes are house-keeping, whereas others have organ-specific functions. The results of the present study will enhance our understanding of the functions of the plasma membrane in various organs, in addition to organ specificity.

## Figures and Tables

**Figure 1 ijms-21-06988-f001:**
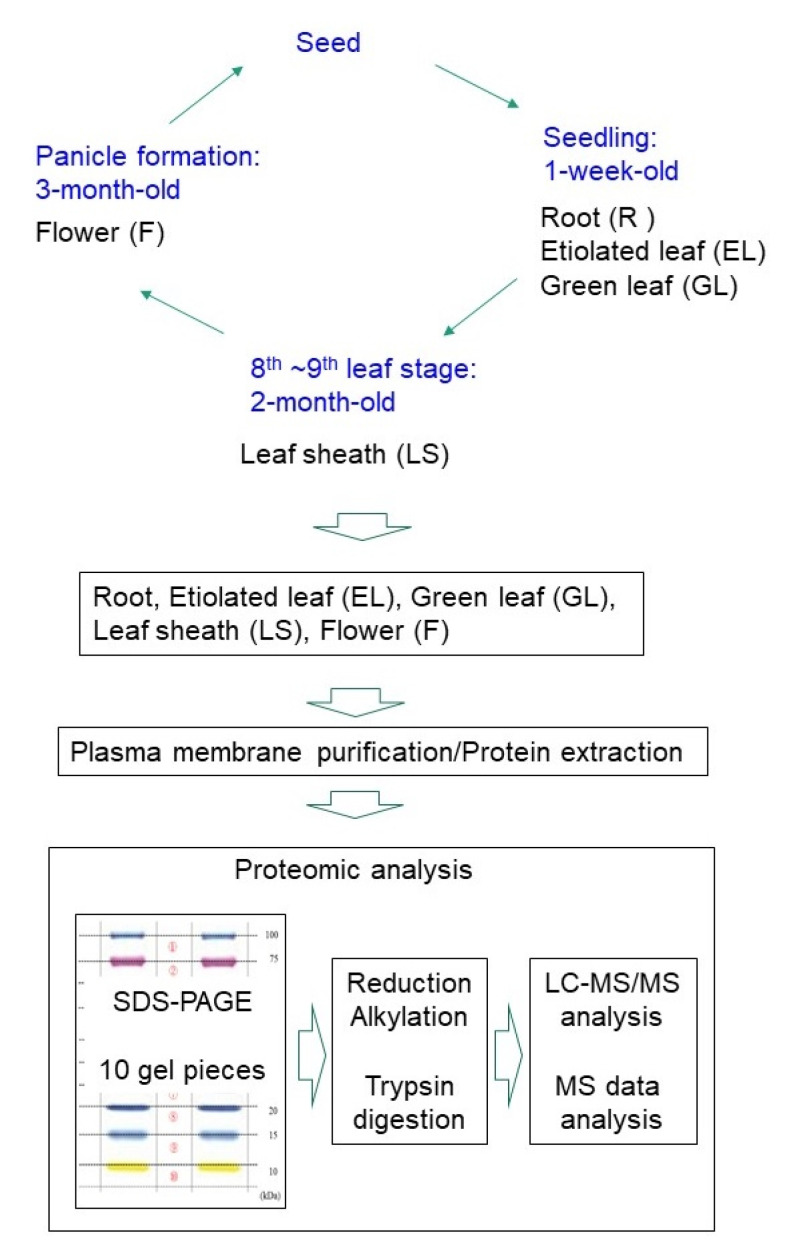
Experimental setup used in this study. Root (R), etiolated leaf (EL), green leaf (GL), leaf sheath (LS), and flower (F) were collected from growing rice plants. Plasma membranes were purified and proteins were extracted from each sample. The extracted proteins were separated by SDS-PAGE and subjected to label-free proteomics analysis.

**Figure 2 ijms-21-06988-f002:**
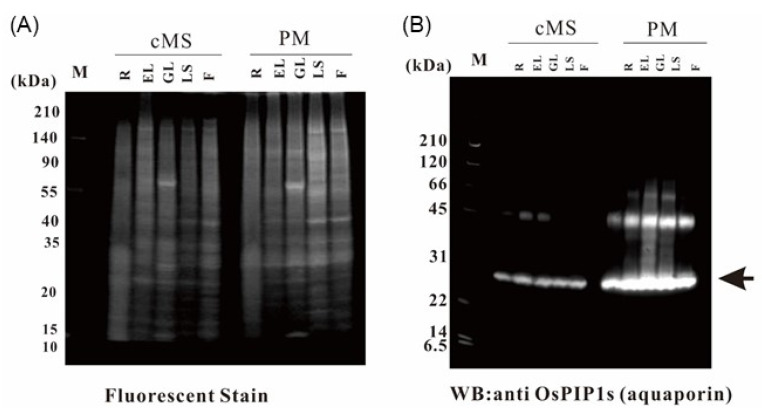
Purification of the plasma membrane fractions. The plasma membrane fractions were purified from the root (R), etiolated leaf (EL), green leaf (GL), leaf sheath (LS), and flower (F) using an aqueous two-polymer phase system. Five micrograms of proteins from crude microsomal fraction and the plasma membrane fractions were separated by SDS-PAGE. (**A**) Oriole fluorescent gel stain. (**B**) Immunoblot with an anti-OsPIP1s antibody targeting the plasma membrane marker aquaporin.

**Figure 3 ijms-21-06988-f003:**
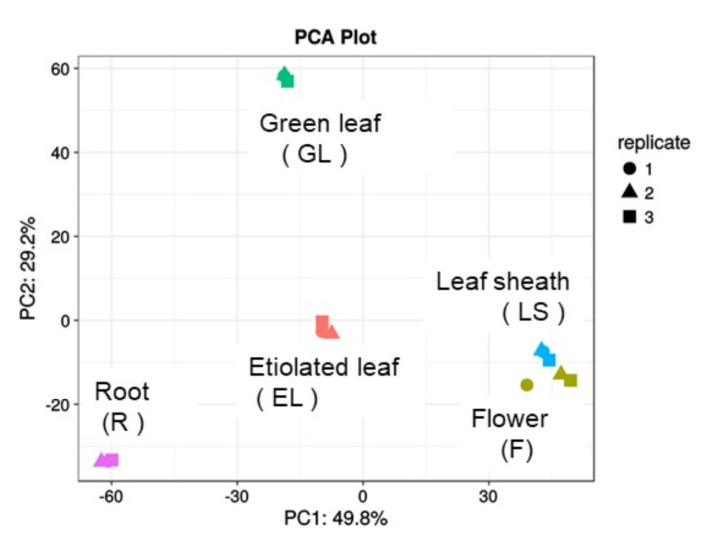
PCA of the total proteome data from the plasma membranes of five organs. We have presented the data of 15 samples, including triplicate samples of five organs, namely, the root (R), etiolated leaf (EL), green leaf (GL), leaf sheath (LS), and flower (F). The raw data were analyzed using MaxQuant to identify and quantify proteins, using in-house standard parameters.

**Figure 4 ijms-21-06988-f004:**
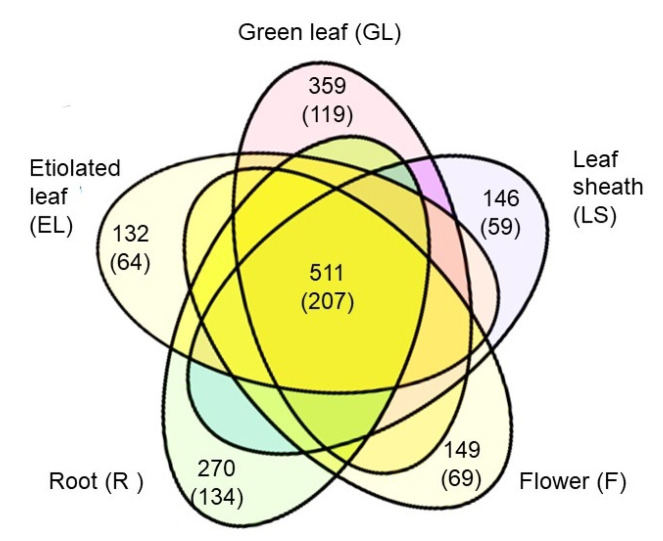
Venn diagram showing the overlap of differentially accumulated proteins in the plasma membrane of the root (R), etiolated leaf (EL), green leaf (GL), leaf sheath (LS), and flower (F). The number of proteins was calculated from the data in Appendix A. The numbers in parentheses indicate the number of proteins assigned to plasma membrane localization, determined using UniProtKB, as listed in Appendix A.

**Figure 5 ijms-21-06988-f005:**
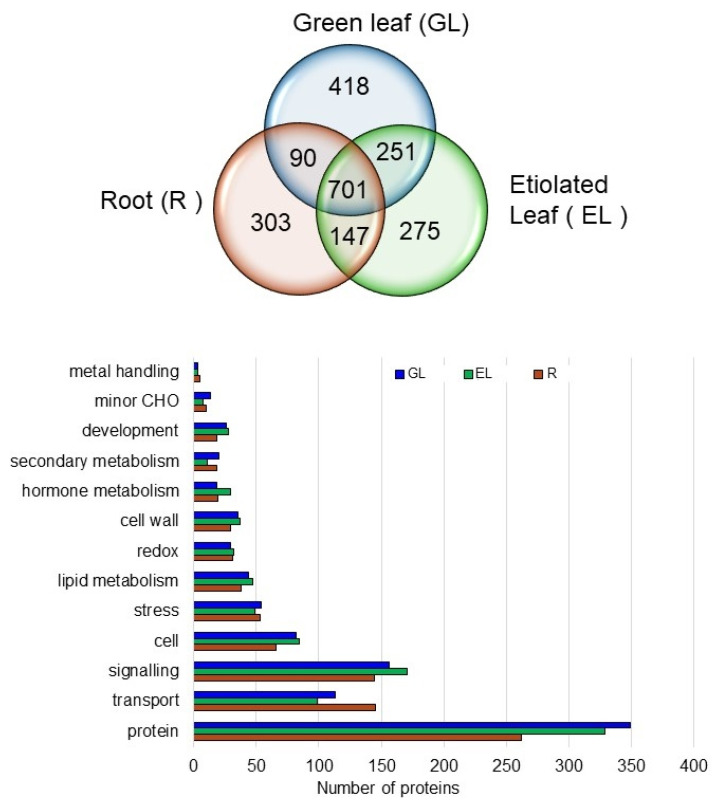
Functional protein categories and Venn diagram showing the number of common and organ-specific proteins in the plasma membrane fractions of the root (R), greening leaf (GL), and etiolated leaf (EL) in 1-week-old rice plants. The plasma membrane proteins were functionally categorized using the MapMan bin codes. Category “cell” contains cell organization, cell division, cell cycle, and vesicle transport. Category “protein” comprises protein degradation/synthesis/targeting/PTM/amino acid activation/folding.

**Figure 6 ijms-21-06988-f006:**
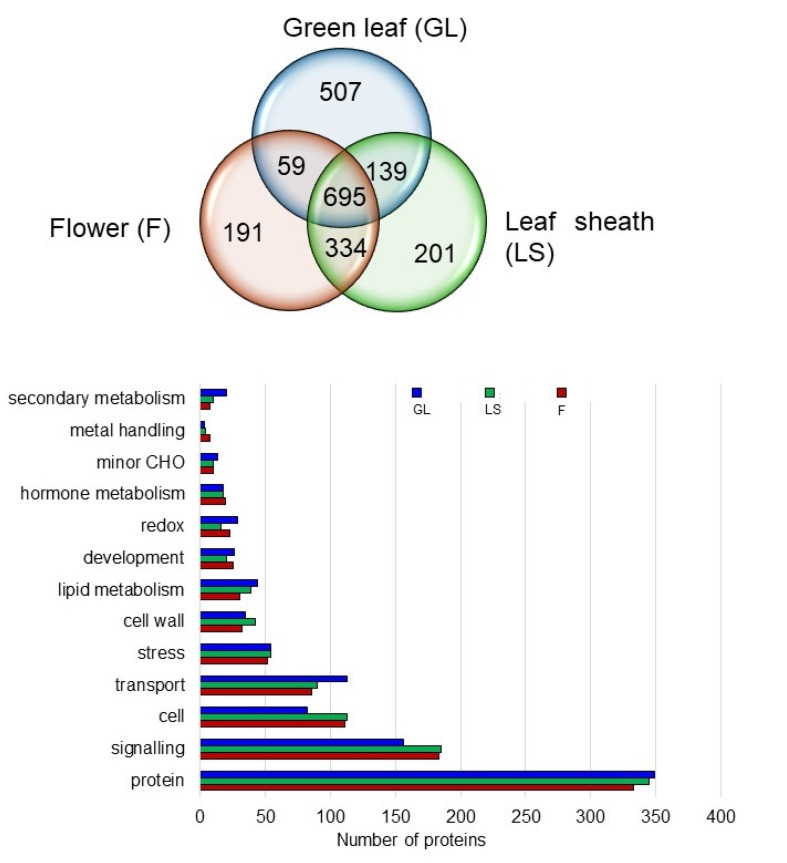
Functional categories and Venn diagram showing the number of common and organ-specific proteins in the plasma membrane fractions of the leaf sheath (LS), flower (F), and green leaf (GL) in rice. The plasma membrane proteins were functionally categorized using the MapMan bin codes. Category “cell” contains cell organization, cell division, cell cycle, and vesicle transport. Category “protein” comprises protein degradation/synthesis/targeting/PTM/amino acid activation/folding.

**Figure 7 ijms-21-06988-f007:**
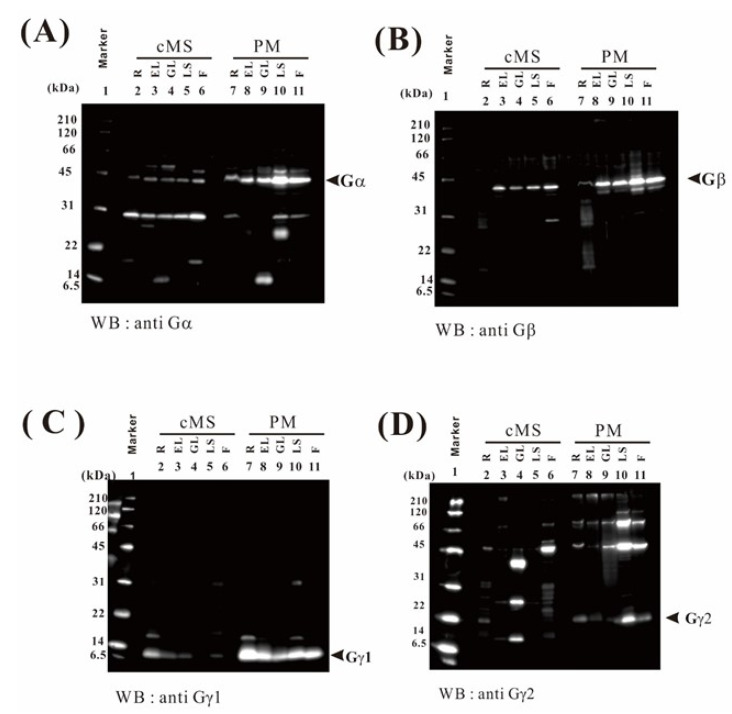
Immunoblot analysis of the Gα, Gβ, Gγ1, and Gγ2 subunits in rice. Proteins (10 μg) in crude microsomal fraction (cMS) and plasma membrane fractions (PM) of the root (R), etiolated leaf (EL), green leaf (GL), leaf sheath (LS), and flower (F) were separated by SDS-PAGE and cross-reacted with the anti-Gα antibody (**A**), anti-Gβ antibody (**B**), anti-Gγ1antibody (**C**), and anti-Gγ2 antibody (**D**). The Gα, Gβ, Gγ1, and Gγ2 subunits are indicated with arrows.

**Figure 8 ijms-21-06988-f008:**
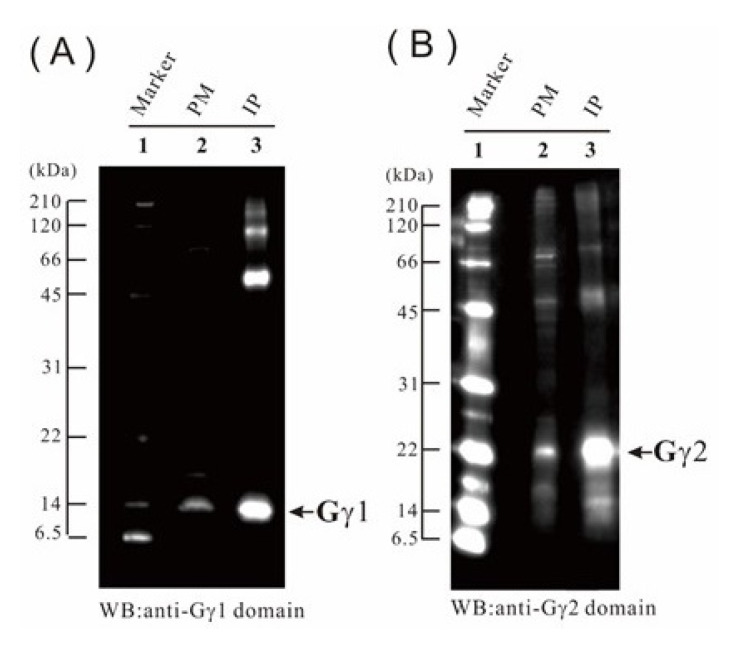
IP of the Gγ1 and Gγ2 subunits in the etiolated leaf. (**A**) IP of Gγ1 from the solubilized plasma membrane protein. Lane 1, molecular weight markers; lane 2, 10 μg of plasma membrane protein; lane 3, IP product of solubilized plasma membrane proteins. Immunoblot analysis was carried out using an anti-Gγ1 antibody. (**B**) IP of Gγ2 from solubilized plasma membrane protein. Lane 1, molecular weight markers; lane 2, 10 μg of plasma membrane protein; lane 3, IP product of solubilized plasma membrane proteins. Immunoblot analysis was carried out using the anti-Gγ2 antibody.

**Figure 9 ijms-21-06988-f009:**
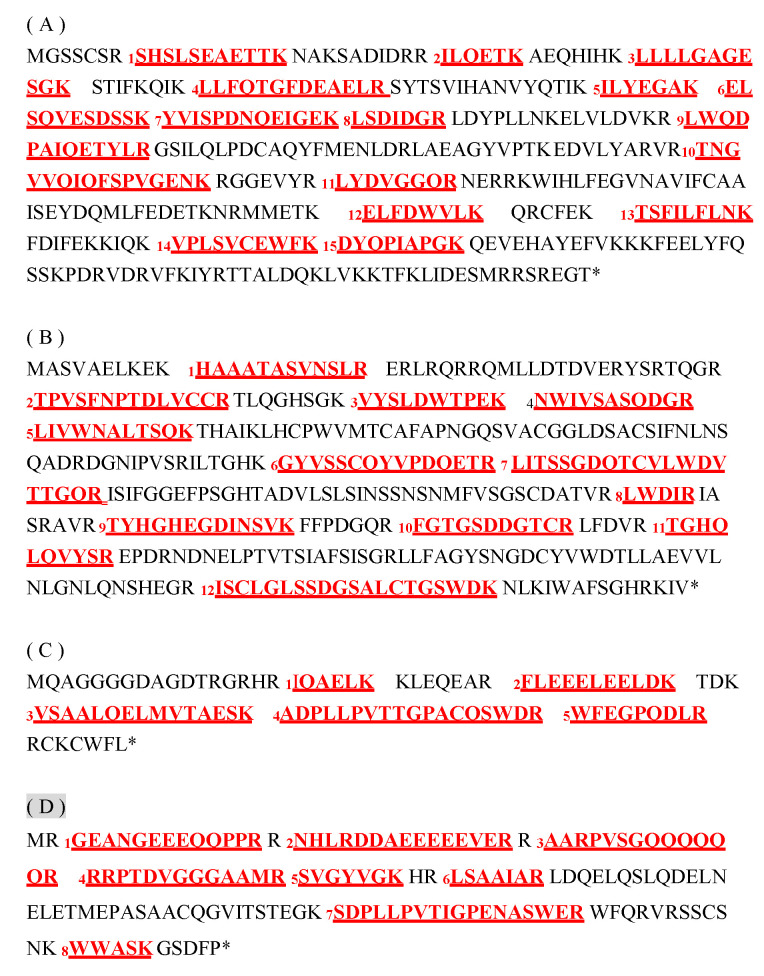
LC-MS/MS analysis of the Gα, Gβ, Gγ1, and Gγ2 subunits in rice. (**A**) Fifteen peptides (*p* < 0.05) produced from trypsin-digested Gα in the plasma membrane fraction are numbered and underlined in the full-length Gα amino acid sequence. These peptides are listed in Appendix A. (**B**) Twelve peptides (*p* < 0.05) produced from trypsin-digested Gβ in the plasma membrane fraction are numbered and underlined in the full-length Gβ amino acid sequence. These peptides are listed in Appendix A. (**C**) Five peptides (*p* < 0.05) produced from trypsin-digested Gγ1 in the IP products are numbered and underlined in the full-length Gγ1 amino acid sequence. These peptides are listed in Appendix A. (**D**) Eight peptides (*p* < 0.05) produced from trypsin-digested Gγ2 in the IP products are numbered and underlined in the full-length Gγ2 amino acid sequences. These peptides are listed in Appendix A.

**Table 1 ijms-21-06988-t001:** Number of proteins, which are grouped into category “transport” using the MapMan analysis, among plasma membrane proteins of the root (R), etiolated leaf (EL), and green leaf (GL).

MapMan Bin Code	MapMan Description	R/EL/GL	R/EL	R/GL	EL/GL	R	EL	GL
34.1	p- and v-ATPases	15	0	2	2	5	2	2
34.2	Sugars	3	2	2	1	3	1	3
34.3	H+ transporting pyrophosphatase/amino acids	3	1	3	0	3	3	3
34.4	Nitrate	0	1	0	0	2	0	2
34.5	Ammonium	1	0	0	0	4	0	0
34.6	Sulfate	0	0	0	0	3	2	2
34.7	Phosphate	0	0	1	1	2	1	1
34.8	Transport. Metabolite transporters at the envelope membrane	0	0	0	0	0	0	2
34.9	Metabolite transporters at the mitochondrial membrane	0	2	0	0	0	0	1
34.12	Metal	0	2	0	0	6	0	1
34.13	Peptides and oligopeptides	2	1	1	1	6	0	2
34.14	Unspecified cations	3	0	0	0	3	0	1
34.15	Potassium	1	1	1	1	1	0	1
34.16	ABC transporters and multidrug resistance systems	7	5	5	2	7	5	5
34.18	Unspecified anions	0	0	0	0	2	0	1
34.19	Major intrinsic proteins (PIP etc.).	7	0	2	3	7	1	1
34.21	Calcium	0	2	0	1	0	0	3
34.99	Misc.	8	1	1	2	5	4	1
Subtotal	□	50	18	18	14	59	19	32

Accessions for individual proteins are shown in Appendix A.

**Table 2 ijms-21-06988-t002:** Number of proteins, which are grouped into category “cell” using the MapMan analysis, in the flower (F), leaf sheath (LS), and green leaf (GL).

MapMan Bin Code	MapMan Description	F/LS/GL	F/LS	F/GL	LS/GL	F	LS	GL
31.1	cell. organization	24	23	0	2	11	6	15
31.2	cell.division	3	1	1	0	0	2	0
31.3	cell.cycle	2	2	0	1	1	1	1
31.4	cell.vesicle transport	16	17	2	5	8	8	10
Subtotal	□	45	43	3	8	20	17	26

Accessions for individual proteins are shown in Appendix A.

**Table 3 ijms-21-06988-t003:** Number of proteins, which are grouped into category “signaling” using the MapMan analysis, in the flower (F), leaf sheath (LS), and green leaf (GL).

MapMan Bin Code	MapMan Description	F/LS/GL	F/LS	F/GL	LS/GL	F	LS	GL
30.1	signaling. in sugar and nutrient physiology	0	0	0	0	0	0	1
30.2	receptor kinases	36	24	2	6	16	11	27
30.3	calcium	17	9	1	1	2	4	5
30.4	phosphoinositide	5	0	0	0	0	3	2
30.5	G-proteins	33	10	1	3	8	6	2
30.6	MAP kinases	0	2	0	0	3	1	1
30.7	14-3-3 proteins	6	0	0	0	1	0	0
30.9	lipids	0	0	0	2	1	0	0
30.11	light	2	1	0	0	2	3	4
30.99	unspecified	0	1	0	0	0	0	0
Subtotal	□	99	47	4	12	33	28	42

Accessions for individual proteins are shown in Appendix A.

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
