# Peer review of "Proteomics Analysis of Plasma Membrane Fractions of the Root, Leaf, and Flower of Rice"

_ijms, 2020, doi:10.3390/ijms21196988_

Round 1
Reviewer 1 Report
The authors here compared the proteomics of root, etiolated seedling, green leaf, leaf sheath, and flower. Proteomics comparison of different rice tissues is very informative to rice or other plant researchers. The authors found differences of plasma membrane proteins of different tissues. Here I would ask what is the novel point or unexpected from their finding? Furthermore, I would ask the authors about more details of sampling, e.g. the absolute amount of different samples, the status of flower (better to have a picture) etc. There are also published proteomic study of plasma membrane proteins of rice, e.g. from roots, the comparison of current data to the published data is also of importance.
Other points are:
Line 66-68. “The results of the present study could not only enhance our understanding of the function of the plasma membrane in various organs but also the characterization of organ specificity in plasma membranes in various organs.” It sounds too general and a bit confusing. Please go to more detail and describe the exact most exciting findings of this study.
Line 76-78. “Sample purity was evaluated by immunoblotting with an antibody against the plasma membrane marker protein aquaporin (OsPIP1s). Aquaporin concentrations in the plasma membrane fractions were compared to those in the crude microsomal fractions (Figure 2).” What conclusion is it here? How pure is the PM? Could this be determined without marker for membranes other than PM?
Line 128-130, “The proteins in these three organs were functionally categorized into transport, signaling, cell, stress, lipid metabolism, redox, cell wall, hormone metabolism, secondary metabolism, and development, using MapMan bin codes.” What is the meaning of cell here? Isn’t it every protein related to cell?
In Figure 5, What is the meaning of “protein” in the last column?
Line 148, “ethiolated” to “etiolated”.
Figure 9 is repeated 2 times.
In supplement Table S11, Table (C) and (D), “ethioleted” should be “etiolated”. Table (D), “Fragmant” should be “Fragment”
In supplement Table S3. “Proteins identified plasma-membrane fractions of rice green leaf.” A in or from is missing between “identified” and “plasma-membrane”.
In supplement Table S6. “Commomly-accumulated-proteins” should be “Commonly-accumulated proteins”.
In supplement Table S8. “Transport regulated proteins” should be “Transport-related” or “Transport-regulating”.
In Table 3 and supplement Table S10. There are “Signaling” and “Signalling”. Please unify.
Author Response
Responses
Reviewer 1:
1-1. The authors here compared the proteomics of root, etiolated seedling, green leaf, leaf sheath, and flower. Proteomics comparison of different rice tissues is very informative to rice or other plant researchers. The authors found differences of plasma membrane proteins of different tissues. Here I would ask what is the novel point or unexpected from their finding?
(Response)
We assume that question 1-1 may be related to question 1-4. We have added the principle component analysis (PCA) results of the proteome data (Fig. 3). The results indicated that each organ has characteristic plasma membrane proteins.
Regarding the novelty of our study, we performed western blotting, by focusing on the heterotrimeric G protein subunits. There are 1 Gα, 1 Gβ, and 5 Gγ genes in rice. The western blotting analysis revealed the presence of the Gα and Gβ subunits; however, Gγ1 and Gγ2 were not present. We showed that Gγ1 and Gγ2 can be detected after concentration by immunoprecipitation. The results suggest that the amount of Gγ1 and Gγ2 in the plasma membrane may be negligible, and therefore, they could not be detected by LC-MS/MS. We have added the following information in the last paragraph of the Discussion.
“Finally, the PCA results of the total plasma membrane proteome data of the five organs indicated that each organ has characteristic plasma membrane proteins. Western blotting analysis, focusing on the heterotrimeric G protein subunits, revealed the presence of the Gα and Gβ subunits. Our proteome data of the five organs will help future plant science research.”
1-2. Furthermore, I would ask the authors about more details of sampling, e.g. the absolute amount of different samples, the status of flower (better to have a picture) etc.
(Response)
Green leaf, etiolated leaf, and root were sampled from plants grown in a plant chamber in our lab. Leaf sheath and flower sampling was slightly difficult. In mid June, approximately 2000 tillers were harvested from the paddy field and dissected in our lab. We used the youngest leaf in the tiller. The length of leaf sheath of the youngest leaf varied from 0 to 20 cm. We collected 1–5-cm long leaf sheath, which accounted for approximately 10% of the dissected tillers. The weight of 1–5-cm long leaf sheath obtained from 200 tillers was approximately 4 g fresh weight. From 4 g of leaf sheath, 200–300 μg of plasma membrane was obtained. The sampling of flower was similar to that of leaf sheath. In mid July, approximately 2000 tillers were harvested from the paddy fields and dissected in the lab. The tillers that contained 1–5-cm long flowers accounted for approximately 10% of the total tillers harvested; 200 tillers were used. The weight of 1–5-cm long flowers from 200 tillers was approximately 4 g fresh weight. From 4 g of flowers, 200–300 μg of plasma membrane was obtained. We have added following information in the Materials and methods:
“Approximately 2000 tillers were harvested from the paddy field and dissected in our lab each day. The leaf sheath length of the youngest leaf of the tiller varied from 0 to 20 cm. We collected leaf sheath of length 1–5 cm, and it accounted for approximately 10% of the dissected tillers. The weight of 1–5-cm long leaf sheath obtained from 200 tillers was approximately 4 g fresh weight. From 4 g of leaf sheath, 200–300 μg of plasma membrane was obtained. The sampling of flower was similar to that of leaf sheath. Approximately 2000 tillers were harvested from the paddy field and dissected in the lab each day. The tillers that contained 1–5 cm flowers accounted for approximately 10% of the total tillers harvested; 200 tillers were used. The weight of 1–5-cm long flowers from 200 tillers was approximately 4 g fresh weight. From 4 g of flowers, 200–300 μg of plasma membrane was obtained. These extracts were used as samples for biological triplicates (Figure 1).”
1-3. There are also published proteomic study of plasma membrane proteins of rice, e.g. from roots, the comparison of current data to the published data is also of importance. ?
(Response)
Per your suggestion, we identified the relevant publication on PubMed and revised the manuscript accordingly.
Roth R, Chiapello M, Montero H, Gehrig P, Grossmann J, O'Holleran K, Hartken D, Walters F, Yang SY, Hillmer S, Schumacher K, Bowden S, Craze M, Wallington EJ, Miyao A, Sawers R, Martinoia E, Paszkowski U. A rice Serine/Threonine receptor-like kinase regulates arbuscular mycorrhizal symbiosis at the peri-arbuscular membrane. Nat Commun. 201, 9(1):4677.
(Discussion)
“Roth et al. [24] purified the plasma membrane and identified 2036 proteins from untreated rice and 3550 proteins including 591 fungal proteins from inoculated rice as rice membrane proteins. Using these proteins, peri-arbuscular membrane-specific arbuscular receptor-like kinase 1 for sustained arbuscular mycorrhizal symbiosis was detected. Although the number of plasma membrane proteins identified in rice root in our study was low, many proteins were the same as plasma membrane proteins identified by Roth et al. [24]. In rice leaves, 3,049 proteins have been previously identified [20], 1,172 of which exist only in the plasma membrane fraction and 1,877 in the cytosolic fraction. In the present study, 1,369 proteins were identified as plasma membrane proteins in the leaf sheath of rice. In addition, proteomic studies on the microdomain structure of plasma membrane in oat, rye [22], and rice [25] could enhance our understanding of plasma membrane functions in plants.”
1-4. Line 66-68. “The results of the present study could not only enhance our understanding of the function of the plasma membrane in various organs but also the characterization of organ specificity in plasma membranes in various organs.” It sounds too general and a bit confusing. Please go to more detail and describe the exact most exciting findings of this study.
(Response)
We thank you for your suggestion, which may be related to question 1-1. We have revised the relevant sentences and moved them from the Introduction to the last paragraph of the Discussion.
The following sentences have been deleted in the Introduction:
The results of the present study could not only enhance our understanding of the function of the plasma membrane in various organs but also the characterization of organ specificity in plasma membranes in various organs.
The following sentences have been added in the Discussion:
“Finally, the PCA results of the total plasma membrane proteome data of the five organs indicated that each organ has characteristic plasma membrane proteins. Western blotting analysis, focusing on the heterotrimeric G protein subunits, revealed the presence of the Gα and Gβ subunits. Our proteome data of the five organs will help future plant science research.”
1-5. Line 76-78. “Sample purity was evaluated by immunoblotting with an antibody against the plasma membrane marker protein aquaporin (OsPIP1s). Aquaporin concentrations in the plasma membrane fractions were compared to those in the crude microsomal fractions (Figure 2).” What conclusion is it here? How pure is the PM? Could this be determined without marker for membranes other than PM?
(Response)
We apologize for our overstatement regarding “sample purity.”
We have deleted the following sentence:
Sample purity was evaluated by immunoblotting with an antibody against the plasma membrane marker protein aquaporin (OsPIP1s).
Furthermore, we have revised the relevant sentence as follows:
“To determine how the proportion of plasma membrane increased in the membrane fraction, immunoblotting with an antibody against the plasma membrane marker protein aquaporin (OsPIP1s) was carried out.”
1-6. Line 128-130, “The proteins in these three organs were functionally categorized into transport, signaling, cell, stress, lipid metabolism, redox, cell wall, hormone metabolism, secondary metabolism, and development, using MapMan bin codes.” What is the meaning of cell here? Isn’t it every protein related to cell?
(Response)
Category “Cell” in the MapMan bin codes contains organization, division, cycle, and vesicle transport. We have added the following sentence in the legend of Figures 5 and 6.
“Category “cell” contains cell organization, cell division, cell cycle, and vesicle transport.”
1-7. In Figure 5, What is the meaning of “protein” in the last column?
(Response) 
We thank you for your comment. It is a category of the MapMan bin codes. We have added the following sentence in the legends of Figures 5 and 6.
“Category “protein” comprises protein degradation/synthesis/targeting/PTM/amino acid activation/folding.”
1-8. Line 148, “ethiolated” to “etiolated”.
(Response)
We apologize for our negligence; we have changed “ethioleted” to “etiolated.”
1-9. Figure 9 is repeated 2 times.
(Response)
We apologize for our negligence; we have made the necessary change in the manuscript.
1-10. In supplement Table S11, Table (C) and (D), “ethioleted” should be “etiolated”. Table (D), “Fragmant” should be “Fragment”
(Response)
We apologize for the oversights. We have changed “ethioleted” to “etiolated” and “Fragmant” to “Fragment.”
1-11. In supplement Table S3. “Proteins identified plasma-membrane fractions of rice green leaf.” A in or from is missing between “identified” and “plasma-membrane”.
(Response)
We apologize for our negligence; we have made the necessary change in the manuscript.
1-12. In supplement Table S6. “Commomly-accumulated-proteins” should be “Commonly-accumulated proteins”. ?
(Response)
We apologize for the oversight. We have changed “Commomly-accumulated-proteins” to “Commonly-accumulated proteins.”
1-13. In supplement Table S8. “Transport regulated proteins” should be “Transport-related” or “Transport-regulating”.
(Response)
We have changed “Transport regulated proteins” to “Transport-regulated proteins.”
1-14. In Table 3 and supplement Table S10.? There are “Signaling” and “Signalling”. Please unify.
(Response)
We apologize for our negligence; we have used “signaling” throughout the manuscript.
Reviewer 2 Report
A well written article which is easy to understand. The study employed the reliable aqueous tow-phase partitioning technique to purify plasma membranes. Then the proteomic analysis of these membranes from different organs generated extensive and useful data on their protein composition.This is a good paper and my opinion is that it should be published after very minor revision.
My only point is that there are some small typos in the manuscripts eg
Ln 69/70 add of ie characterisation of
ln 75 five not fiver
Ln 439 flower not flow
The authors' should read through the manuscript carefully and find and correct any other such mistakes.
Author Response
Reviewer 2:
2-1. Ln 69/70 add of ie characterisation of
(Response)
We apologize for our negligence; we have made the necessary change in the manuscript.
2-2. ln 75 five not fiver
(Response)
We apologize for our negligence; we have changed “fiver” to “five.”
2-3. Ln 439 flower not flow
(Response)
We apologize for our negligence; we have changed “flow” to “flower.”
2-4. The authors' should read through the manuscript carefully and find and correct any other such mistakes.
(Response)
We thank you for your suggestion. We rechecked the manuscript and made changes where necessary.
Reviewer 3 Report
The manuscript submitted by Iwasaki et al. reports a proteomic analysis of plasma membrane fractions extracted from different plant tissues of Oryza sativa L. cultivar Nipponbare. The analysis provides information about the relationship between protein composition and organ specificty in this cultivar of rice. Authors show a classification of identified proteins. The text and Figures are clear and well presented. The methodology is also well described. Discussion shows the results in the context of other related studies in different plant species.
Before publication authors should amend some details.
Minor comments:
- The title should include the plant specie studied.
- Lines 98, 108, 110, 113: for clarity and help the reader writte the numbers between parenthesis after the corresponding organ/tissue.
- Lines 125, 126, 153, 181: revise the number of proteins. They do not correspond well with data of figures.
- Line 181, authors write "In category signaling, a total of 265 proteins were categorized into....", however in Figure 6 signlallig proteins are lower than 200.
Author Response
Reviewer 3
3-1. The title should include the plant specie studied.
(Response)
We revised the title, as following ;
“Proteomics Analysis of Plasma Membrane Fractions of the Root, Leaf, and Flower of Rice”.
3-2. Lines 98, 108, 110, 113: for clarity and help the reader writte the numbers between parenthesis after the corresponding organ/tissue.
(Response)
We have change the sentences (from lane 98~113) according to reviewer suggestion;
In total, more than 1,000 proteins were identified in the plasma membrane fractions of the root(1,181), etiolated leaf (1,314), green leaf (1,400), leaf sheath (1,369), and flower (1,279), respectively (Tables S1–S5).
A Venn diagram was generated to show the number of common and organ-specific plasma membrane proteins among proteins from the root (1,181), etiolated leaf (1,314), green leaf (1,400), leaf sheath (1,369), and flower (1,279), respectively (Figure 4). In total, 511 proteins were commonly accumulated in all organs evaluated, whereas there were specifically accumulated proteins in the root (270), etiolated leaf (132), green leaf (359), leaf sheath (146), and flower (149), respectively. In the Gene Ontology analysis, among the 511 common proteins, 207 proteins were categorized as plasma membrane proteins (Figure 4 and Table S6). In the UniProtKB analysis, 445 proteins were categorized as plasma membrane proteins specific to the root (134), etiolated seedling (64), green leaf (119), leaf sheath (59), and flower (69), respectively (Table S7),
3-3. Lines 125, 126, 153, 181: revise the number of proteins. They do not correspond well with data of figures.
(Response)
We thank you for your suggestions. We revised sentences.
Lane 125, 126: We added breakdown of 944, as following 944 (418+251+275)
In total, 701 proteins were commonly accumulated in the root, etiolated leaf, and green leaf, whereas 944 (418+251+275) proteins were specifically accumulated in the green and etiolated leaves, and 303 were root specific (Figure 5).
Lane 153; We added Figure number in the sentence.
In total, 695 proteins were commonly accumulated in leaf sheath, flower, and green leaf (Figure 6).
Lane 181;  We revised sentences.
In category “signaling,” 265 proteins (integrated value of subtotal of Table 3) were categorized into the following groups, using MapMan bin codes: sugar/nutrient physiology, receptor kinase, calcium, phosphoinositide, G-protein, MAP kinase, 14-3-3, lipids, light, and specified. The 265 proteins were also grouped into seven groups as described above ( F/LS/GL, F/LS, F/GL, LS/GL, F, LS, GL); there were 99 proteins in F/LS/GL and 47 in F/LS (Table 3).
3-4. Line 181, authors write "In category signaling, a total of 265 proteins were categorized into....", however in Figure 6 signlallig proteins are lower than 200.
(Response)
We thank you for your suggestions. The question (3-4) is related to Question 3-3.
As there are organ-specific proteins, total number of Table 3 become more than number in each organ of Fig. 6.
Round 2
Reviewer 1 Report
The authors have addressed all of the concerns I had with the previous submission. I suggest the paper to be accepted after formatting check and minor spell check.